# Development of an Index for Forest Fire Risk Assessment Considering Hazard Factors and the Hazard-Formative Environment

**Adu Gong** [1,2]**, Zhiqing Huang** [1,2,]*****, Longfei Liu** [3]**, Yuqing Yang** [1,2]**, Wanru Ba** [1,2] **and Haihan Wang** [1,2]

[1] State Key Laboratory of Remote Sensing Science, Faculty of Geographical Science, Beijing Normal University, Beijing 100875, China; gad@bnu.edu.cn (A.G.); ygtq@mail.bnu.edu.cn (Y.Y.); wanruba@mail.bnu.edu.cn (W.B.); whh@mail.bnu.edu.cn (H.W.)

[2] Beijing Engineering Research Center for Global Land Remote Sensing Products, Faculty of Geographical Science, Beijing Normal University, Beijing 100875, China

[3] National Disaster Reduction Center of China, Ministry of Emergency Management of the People's Republic of China, Beijing 100124, China; liulongfei@ndrcc.org.cn

***** Correspondence: huangzhiqing@mail.bnu.edu.cn

**Abstract:** Forest fires are characterized by a rapid and devastating nature, underscoring the practical significance of forest fire risk monitoring. Currently, forest fire risk assessments inadequately account for non-meteorological hazard factors, lack the hazard-formative environment and contextual disaster knowledge for fire occurrence mechanisms. In response, based on MODIS products, we augmented the FFDI (forest fire danger index) with the RDST (regional disaster system theory) and selected various fire risk indicators, including lightning. MOD14 was used for the correlation analysis of fire and its indicators. Through the amalgamation of the analytic hierarchy process (AHP), the entropy method, and the minimal relative entropy theory, we formulated the CFFRI (composite forest fire risk index) and assessed forest fire risks spanning from 2010 to 2019 in Southwest China, which were validated with historical disaster data and MCD64. The findings revealed that the CFFRI yields consistently higher overall fire risk values, with 89% falling within the high-risk category and 11% within the moderate-risk category. In contrast, the FFDI designated 56% of cases as fourth-tier fire risks and 44% as third-tier fire risks. Notably, the CFFRI achieved an accuracy of 85% in its calculated results, while the FFDI attained 76%. These outcomes robustly demonstrate a superior applicability of the CFFRI compared with the traditional FFDI.

**Keywords:** forest fire risk; MOD14 product; MCD64 product; indicator system; regional disaster system theory; hazard assessment

## 1. Introduction

Forest fires are frequent natural disasters in China and around the world, characterized by their sudden onset, significant destructive power, and challenging emergency response efforts. Globally, more than 200,000 forest fires occur each year, resulting in the destruction of millions of hectares of forest, accounting for over 0.1% of the total forest area [1,2]. In recent years, forest fires have erupted frequently worldwide, with numerous significant incidents occurring in 2019 alone [3–9]. China has consistently faced severe forest fire hazards, with a major forest fire having occurred in Liangshan Prefecture, Sichuan Province, on 30 March 2019, resulting in 31 fatalities and the burning of 20 hectares of forest land. A year later, on 30 March another massive forest fire struck the same region, causing 19 casualties, 3 injuries, and affecting an area of approximately 3048 hectares, with direct economic losses reaching nearly a billion yuan [10]. Forest fires not only directly destroy vegetation, resulting in substantial losses, but also indirectly impact the carbon cycling process and distribution patterns within the biosphere, disrupting regional ecosystem equilibrium [2,9,11]. Therefore, conducting pre-disaster warning and risk monitoring

research on forest fires holds significant importance in mitigating disaster losses, protecting forest resources, and maintaining ecological balance [12,13].

Since the 1970s, remote sensing technology enabled the monitoring of regional wildfires based on abundant satellite data. In 1999, NASA successfully launched the first satellite of the Earth Observing System (EOS) program, TERRA, equipped with five Earth-observing sensors, including the Moderate Resolution Imaging Spectroradiometer (MODIS) [14,15]. The MODIS sensor includes a dedicated channel designed for wildfire monitoring [16]. MODIS data are readily accessible, relatively easy to process, and offer high-temporal resolution, making them the most essential remote sensing data for global wildfire research in the past decade [17].

Fuel, such as forests, could cause fires due to various reasons [18,19]. Increasing human activities have put forests in a flammable situation, such as burning straw for the autumn harvest, fireworks for traditional celebrations, and other ignition behaviors. Moreover, forest phenology could directly affect the stocking level, causing seasonal differences in fire cases. It has also been proven that the lower-latitude regions are more prone to risk fire hazards [18]. Facing such uncertainty, vision-based SAG (space–air–ground) remote sensing has become a prevalent strategy for forest fire detection and risk assessment [20].

Based on remote sensing, risk assessments of forest fires adapted to different situations have been developed, and currently, forest fire risk assessment methods can be broadly categorized into three types as follows:

- Forest fire risk assessments based on mathematical and statistical methods.

These assessments analyze and study data characteristics using statistical theories, such as cluster analysis, fuzzy comprehensive evaluation, and the gray system theory, to derive statistical patterns. For instance, Lv et al. developed a forest fire spread GM (1,1) model based on the gray system modeling theory and the ER algorithm directly using MODIS fire point data. However, continuous updates and adjustments to the model are required to avoid the distortion of results [21]. Krishna Prasad Vadrevu, in 2010, employed a combination of the fuzzy set theory and decision algorithms within a Geographic Information System (GIS) framework to analyze forest fire risk in southern India and created fire risk maps [22]. Sirio Modugno and colleagues, in 2016, utilized logistic regression analysis to explore specific positive and negative relationships between forest fires and the wildland–urban interface (WUI) in their vicinity [23]. In 2017, Ryan Lagerquist and his team predicted fire spread days and extreme weather fire risk using self-organizing maps (SOMs) based on the Canadian Fire Weather Index System (CFWIS) [24]. H. Yathish and colleagues, in 2019, applied three fire hazard discrimination methods (logistic regression, multi-criteria decision analysis, and weighted overlay) in India, considering factors like elevation, slope, distance to roads, human activity area, land surface temperature (LST), and the normalized difference vegetation index (NDVI) for fire risk assessment. They found the logistic regression model to have the highest accuracy at 88.89% [25]. Volkan Sevinc et al., in 2019, used a Bayesian network model to predict fire risk factors in southwestern Turkey, highlighting the month as the primary influencing factor on forest fire occurrence, followed closely by the temperature; however, they did not consider the relationship between the temperature and the month [26]. These methods demand high data accuracy and reliability and possess a degree of randomness. However, they incorporated a limited mechanistic understanding of fire occurrence and background knowledge.

- Forest fire risk assessments based on forest fire danger indices.

This category constructs the fire risk assessment index based on the mechanisms of fire occurrence, providing a more realistic reflection of fire occurrence. In 1967, McArthur introduced the fire danger index (FDI) based on meteorological factors, which laid the foundation for this approach [27]. The Canadian Forest Fire Danger Rating System (CFFDRS) is widely used and has been applied in several countries, including parts of the United States, New Zealand, Fiji, Indonesia, and Malaysia [28]. This system calculates multiple flammability and fire behavior indicators based on four meteorological factors and derives

a risk index for quantitatively guiding forest fire management activities [29,30]. In China, the adaptability of the CFFDRS was verified in the Daxing'anling region [31]. In 1995, the Chinese Ministry of Forestry introduced the National Forest Fire Weather Rating Standard, considering five fire risk meteorological factors and employing an expert scoring method. In 2007, the FFDI (forest fire danger index) was optimized and released, undergoing further improvement based on the modified Brong–Davis scheme in 2018, and continues to be used today [32]. However, traditional fire risk indices primarily focus on meteorological and fuel factors, have limited factors for the surrounding environment such as terrain, the water system, and other geographical factors, and offer a relatively macroscopic assessment perspective, making them less adaptable for regional forest fire assessment, as they may not consider all relevant fire impact factors.

- Forest fire risk assessments based on machine learning methods.

In recent years, machine learning methods have gained popularity in fire risk assessment. Support vector machines (SVMs) are commonly used and have demonstrated good performances in fire risk assessment model comparison experiments [33–35]. Ensemble algorithms, like random forest (RF), gradient boosting decision tree (GBDT), and XGBoost, demonstrate better accuracy compared to most individual algorithms and have exhibited strong classification and regression prediction capabilities in various forest fire risk assessment studies [36–39]. Multilayer perceptrons (MLPs) and back propagation neural networks (BPNNs) are also frequently employed in forest fire risk assessment, although their accuracy tends to be lower than the aforementioned methods [33–43]. Deep learning methods, such as recurrent neural networks (RNNs) and convolutional neural networks (CNNs), map complex relationships between fire occurrences and input features through intricate network structures [44–47]. However, machine learning methods require a high quality and quantity of data, necessitate continuous adjustments and optimizations with changing data, involve lengthy and complex training processes, and face challenges in handling diverse geographic conditions. In large-scale predictions, the complexity of these models increases significantly, making them less suitable, and the "black-box" nature of machine learning may lead to weaker reliability [44–47].

In conclusion, considering the advantages and disadvantages of various methods, we have chosen to use the widely adopted fire risk index approach. Building upon the FFDI, we took into account the mechanisms of fire occurrence, analyzed not only the hazard-affected body but also the hazard-formative environment and various hazard factors, including lightning, and constructed a composite forest fire risk index, providing assessment results for the study area from 2010 to 2019. We also incorporated historical disaster data for result validation, thereby offering valuable insights for local forest fire monitoring and prevention efforts.

## 2. Materials and Methods

### 2.1. Sudy Area

The study area encompasses the Liangshan Prefecture and Panzhihua City in Sichuan Province, China (Figure 1). Sichuan Province is a major contributor to forest resources in Southwestern China, with forests occupying 184,000 square kilometers, accounting for 38% of the province's land area and 7.6% of the whole country. From 2010 to 2019, Sichuan Province experienced a cumulative total of 3066 forest fires, contributing to 8.1% of the total forest fires nationwide, which amounted to 37,887 incidents. Facing a high incidence of forest fires, the province's most fire-occurring area is the "Three Prefectures and One City" region, which includes Liangshan Prefecture, Aba Prefecture, Ganzi Prefecture, and Panzhihua City. Among them, Liangshan Prefecture and Panzhihua City have a higher frequency of forest fires, while Ganzi Prefecture and Aba Prefecture suffer more significant losses due to these fires [48]. The study area covers a total area of 67,800 square kilometers and is situated from latitude 26°03′ to 29°18′N and from longitude 100°03′ to 103°52′E, with a subtropical monsoon climate. The topography is characterized by higher

elevations in the northwest and lower elevations in the southeast, with a predominance of mountainous terrain.

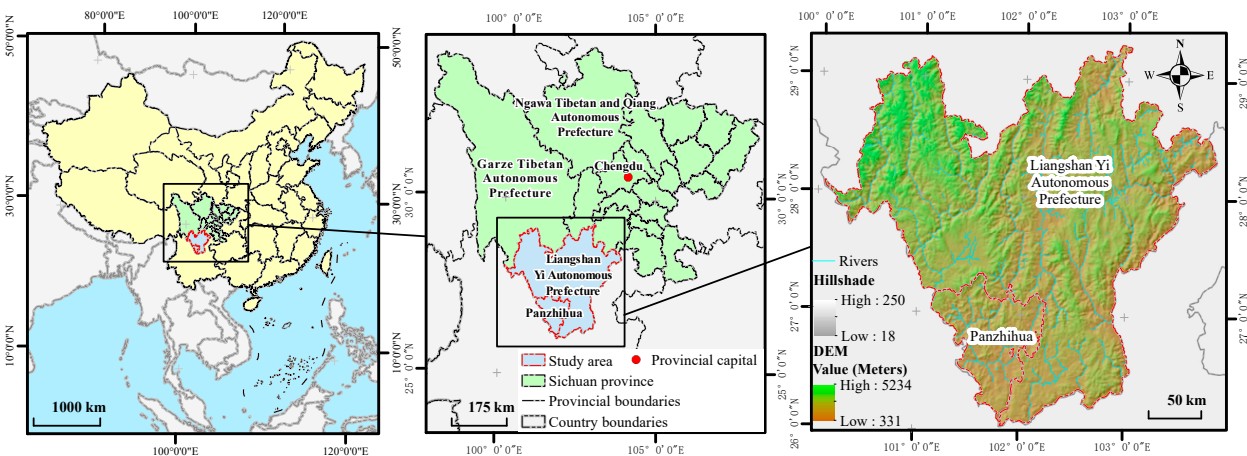

**Figure 1.** The study area.

*2.2. Data Sources*

The data we used in our study were of various types, including meteorological and lightning data, the vegetation category, population density, basic geographic data, historical disaster data, and MODIS products, specifically MOD14 and MCD64. Detailed entries and details for each type of data are shown in Table 1.

**Table 1.** Summary of collected datasets.

| Data Type | | Year | Format | Spatial Resolution | Data Source |
|---|---|---|---|---|---|
| Meteorological data | | 2010–2019 | .csv | - | http://data.cma.cn (accessed on 2 March 2021) |
| Lightning | | 2016–2019 | .xls | - | https://www.scdsjzx.cn/ (accessed on 5 March 2021) |
| Vegetation type | | 2018 | .tiff | 1 km | http://www.resdc.cn (accessed on 14 April 2021) |
| Population density | | 2015 | .tiff | 1 km | http://www.resdc.cn (accessed on 14 April 2021) |
| Basic geographic data | River system | 2019 | .shp | - | https://wiki.openstreetmap.org/wiki/Planet.osm (accessed on 24 February 2021) |
| | Roads | 2015 | .shp | - | |
| | DEM | 2007 | .tiff | 1 km | http://srtm.csi.cgiar.org/srtmdata (accessed on 19 April 2021) |
| | Land cover | 2015 | .shp | - | http://www.resdc.cn (accessed on 11 January 2021) |
| Historical disaster data | Number of fires and disaster damage | 2010–2019 | .xls | - | https://data.stats.gov.cn (accessed on 9 March 2021) |
| | Fire case | 2010–2019 | .xls | - | http://slcyfh.mem.gov.cn (accessed on 11 March 2021) |
| MODIS products | MOD14 | 2010–2019 | HDF | 1 km | https://earthengine.google.com (accessed on 3 April 2021) |
| | MCD64 | 2010–2019 | HDF | 1 km | http://mas.arc.nasa.gov (accessed on 21 January 2021) |

2.2.1. Meteorological Data

The meteorological data used in this study were sourced from the China Meteorological Data Network (http://data.cma.cn) (accessed on 2 March 2021) and specifically from the "China Surface Climate Daily Data Set (V3.0)", covering a total of 23 meteorological stations in the research area and its surrounding regions (Figure 2a). As these data are in CSV format, in order to facilitate their inclusion in the final calculation of the fire hazard index for the study area, along with other raster data types, such as TIFF, they were interpolated using Kriging to generate 1 km gridded data. This dataset included the daily maximum temperature, average 2 min wind speed, minimum relative humidity, and precipitation from 20:00 to 20:00 the next day. After correcting the data, and in accordance

with the Chinese national standard "Meteorological Grades of Forest Fire Danger" (GB/T 36743-2018) [49], we calculated the drought level, precipitation correction coefficient ($C_r$), and snowfall correction coefficient ($C_s$) and incorporated them into the calculation of the FFDI along with wind speed, precipitation, and relative humidity. Ultimately, six meteorological indicators were used in the analysis, excluding precipitation from 20:00 to 20:00 the next day.

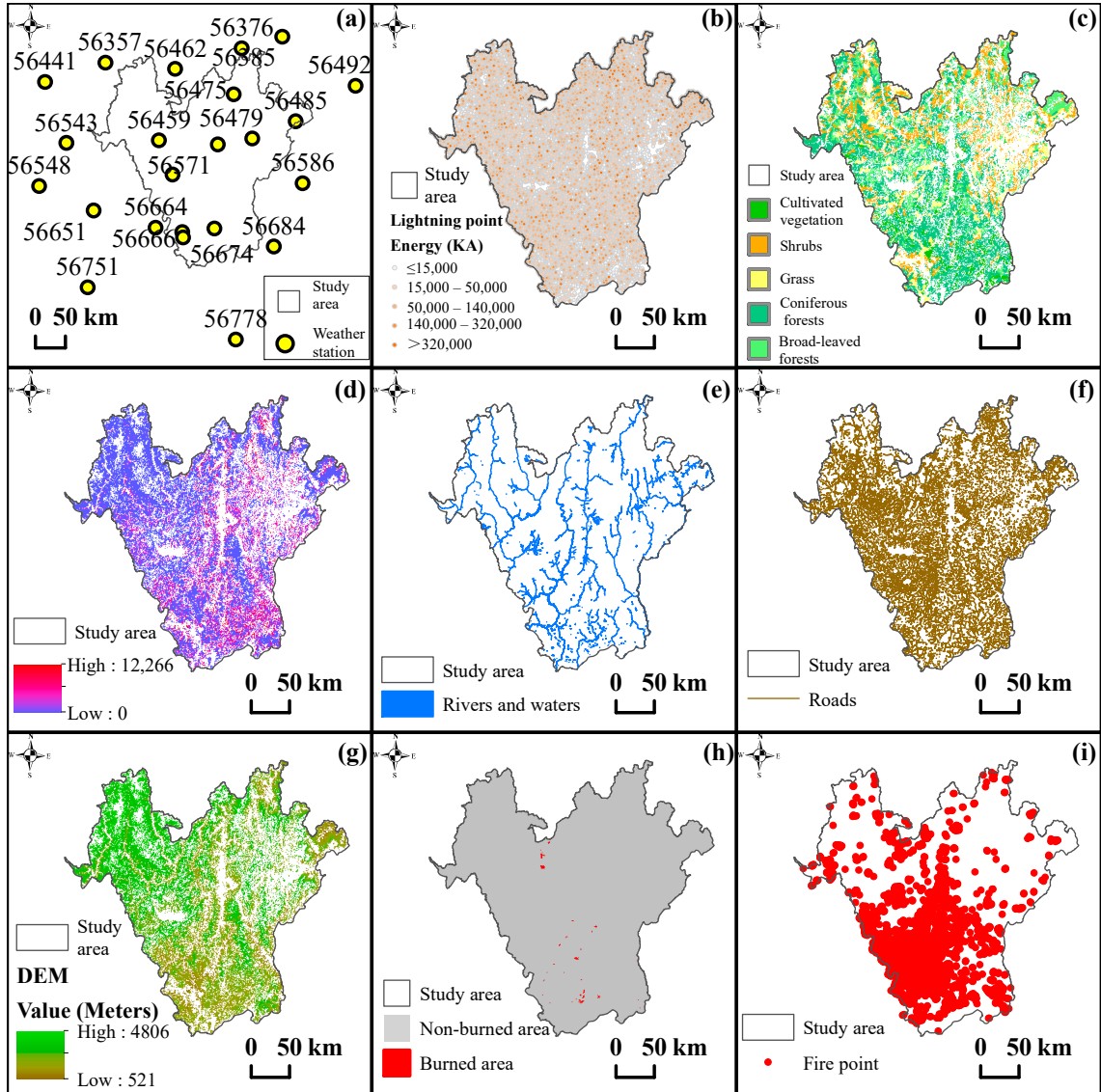

**Figure 2.** Data used in this study: (**a**) weather stations; (**b**) lightning data; (**c**) vegetation data; (**d**) population density; (**e**) river system; (**f**) road data; (**g**) digital elevation model data; (**h**) MCD64A1 burned area data, taking 01/2010 as an example; (**i**) MOD14 fire point data in 2010–2019.

### 2.2.2. Lightning Data

The lightning data, spanning the years from 2016 to 2019, were obtained from statistical records provided by the Sichuan Provincial Big Data Center in China (https://www.scdsjzx.cn/) (accessed on 5 March 2021). These records encompass various attributes, including time, longitude, latitude, intensity, steepness, charge, and energy, which were utilized for the statistical analysis of forest fires triggered by natural factors among the contributing disaster factors. The primary focus of this study was on the attribute of lightning intensity, specifically, the maximum current intensity and impact size of ground discharges from

charged cloud bodies, representing the quantity of lightning strikes (measured in kA). After filtering out anomalous values in the original data and verifying attribute information, a total of 45,000 lightning records were retained. Using ArcMap 10.6 software, the lightning data were correlated with a kilometer grid network covering the study area, resulting in the calculation of lightning intensity per grid unit (data from 2016 and 2017 were combined to represent a single year, totaling three years of lightning data) (Figure 2b); thus, it can participate in risk calculations in the form of a grid.

### 2.2.3. Vegetation Category Data

The vegetation category data were sourced from the Resource and Environmental Science Data Center of the Chinese Academy of Sciences (http://www.resdc.cn) (accessed on 14 April 2021). This dataset comprises information regarding the distribution of 54 vegetation types across 11 vegetation type groups in China. After cropping the data to the study area's extent, a total of five major vegetation categories were obtained within the research area, which includes coniferous forests, broad-leaved forests, cultivated vegetation, shrublands, and grasslands (Figure 2c).

### 2.2.4. Population Density Data

The population density data were obtained from the Resource and Environmental Science Data Center of the Chinese Academy of Sciences (http://www.resdc.cn) (accessed on 14 April 2021). These data have a spatial resolution of 1000 m and represent the population distribution as of the year 2015. These data were utilized to assess the impact of human factors on forest fires within the hazard-formative environment after extracting by the forest vector data for the study area in the year 2015 (Figure 2d).

### 2.2.5. Basic Geographic Data

- The river system and roads.

The water network data for the year 2019 and the road network data for the year 2015, both utilized in this study, were sourced from OpenStreetMap (https://wiki.openstreetmap.org/wiki/Planet.osm) (accessed on 24 February 2021) (Figure 2e,f). The water network data primarily include two attributes: name and length, which were employed to investigate the influence of the distance from rivers on forest fires within the hazard-formative environment. The road network data encompass three main attributes: name, grade, and length, and were used to explore the impact of roading density per kilometer grid on forest fires within the hazard-formative environment.

- The DEM (digital elevation model).

The DEM data were sourced from SRTM (http://srtm.csi.cgiar.org/srtmdata) (accessed on 19 April 2021). The current dataset (Version 4) was generated based on NASA's release of the completed level 3 arc-second SRTM data. The original data from the year 2000 had data gaps, as the radar data lacked sufficient contrast to extract information about water bodies, snow-covered areas, and mountainous elevations. The CGIAR-CSI SRTM dataset underwent post-processing of the NASA data, "filling in" these data gaps in 2007, resulting in a seamless, global coverage of the elevation data (Figure 2g). These data were utilized in this study to investigate the impact of different altitudes on forest fires within the hazard-formative environment.

### 2.2.6. Land Cover Data

The land use data were obtained from the Chinese Academy of Sciences Resource and Environmental Science Data Center product (http://www.resdc.cn) (accessed on 11 January 2021). This dataset was based on Landsat remote sensing images and was developed through manual visual interpretation and field measurements, verified, and constructed into a 1:100,000 scale national Land Use and Land Cover Change (LUCC) thematic database. For this study, the primary data source was the 2015 forest vector data,

combined with the administrative boundaries of Liangshan Prefecture and Panzhihua City. From this combined dataset, forested areas within the study area were extracted to define the final research scope implemented in the calculations.

### 2.2.7. MODIS Products

- MCD64 burned area data.

MCD64 data were obtained from NASA (http://mas.arc.nasa.gov) (accessed on 21 January 2021) in the form of MCD64 MODIS level 3 products, known for their completeness and consistency. They have a spatial resolution of 500 m and a temporal resolution of 1 month. These data include attributes such as BurnDate, uncertainty, quality, FirstDate, and LastDate. We used the MODIS Reprojection Tool Swath to project the original data from sinusoidal projections to the Albert projection, resampling it to 1000 m. The data format was converted from HDF-EOS into TIFF. It underwent further processing, including cropping for the study area and extracting single-band imagery, to facilitate its use in validating the results of the FFDI calculations. The figure shows the burned area data for the study area in January 2010 (Figure 2h).

- MOD14 fire point data

MOD14 data were sourced from the Google Earth Engine (https://earthengine.google.com) (accessed on 3 April 2021). They have a spatial resolution of 1 km and a temporal resolution of 1 day. These data primarily include surface fire masks and quality information, comprising details such as the location of surface fires, radiative energy from fire points, and confidence levels. The data used in this study were the accumulation of daily fire point data for Liangshan Prefecture and Panzhihua City over a 10-year period from 2010 to 2019. They were employed to investigate the relationship between fire points and various influencing factors (Figure 2i).

### 2.2.8. Historical Disaster Data

Part of the historical disaster data used in this study was sourced from the National Bureau of Statistics (https://data.stats.gov.cn) (accessed on 9 March 2021). These data include information on the total number of forest fire occurrences in Sichuan Province for each year from 2010 to 2019, the number of forest fires of different types and levels, the affected forest area, and the number of casualties. Some of the historical disaster data came from the China Forest and Grassland Fire Prevention Network (http://slcyfh.mem.gov.cn) (accessed on 9 March 2021), which includes a total of 35 forest fire incidents in Liangshan Prefecture and Panzhihua City from 2010 to 2019, with a total burned area of 934 hectares. Among the identified causes of these fires, two-thirds were attributed to human factors, such as villagers burning fields and smoking, while one-third resulted from lightning and electrical failures. We confirmed coordinates based on the names of the affected villages and used this information, along with the MCD64 burned area data, for validating the results of the FFDI.

### 2.3. Methods

The overall technical route comprises three main parts (Figure 3). The first part involves the utilization of various datasets in this research. In the second part, based on the regional system disaster theory, hazard factors, the hazard-affected body, and the hazard-formative environment were considered. Various forest fire indicators were selected, and the FFDI was calculated based on them. This part aimed to explore the relationships between indicators and forest fires, forming a foundation for the forest fire risk assessment index system and providing a quantification reference. In the third part, the composite forest fire risk index (CFFRI) was constructed based on the AHP (analytic hierarchy process) and the entropy method. It was then applied to produce a fire risk map, which was validated using the historical disaster data and MCD64 burnt area data and compared with the traditional FFDI results.

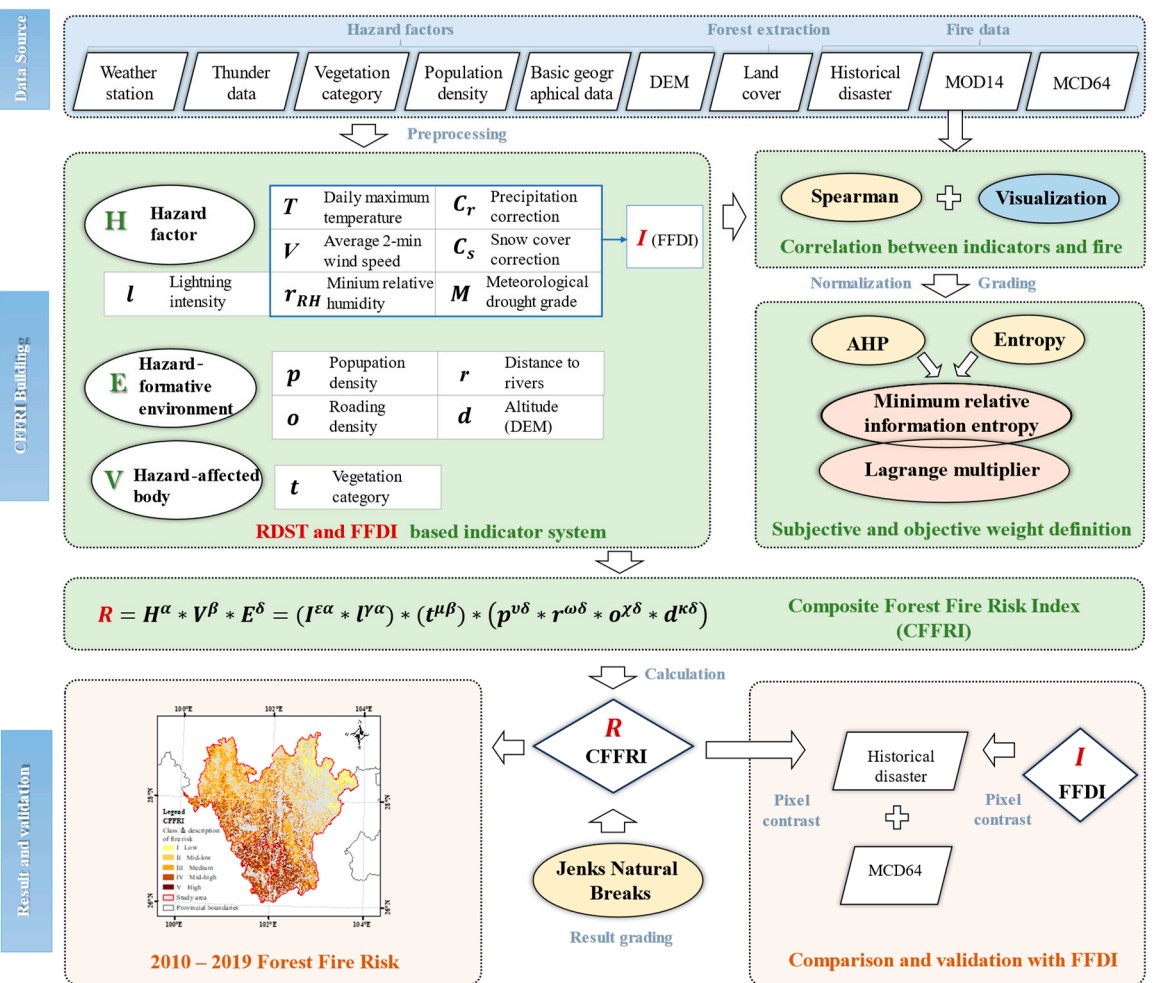

**Figure 3.** The technical route.

### 2.3.1. The Regional Disaster System Theory

Both domestic and international research institutions and scholars have proposed a series of concepts, assessment frameworks, and expressions regarding natural disaster risk, which can be broadly categorized into three aspects [50–52]: (1) from the perspective of risk itself, it defines risk as losses under certain probability conditions; (2) from the perspective of hazard factors, it views disaster risk as the maximum probability that these factors can generate or occur; (3) from the viewpoint of the disaster risk system theory, it considers that disaster risk is mainly the result of the combined effects of the hazard of hazard factors, the exposure of vulnerable objects, and vulnerability. It places a high emphasis on the vulnerability of human society and the economy and its influence on the formation of disasters [53].

Shi, based on the natural disaster risk index model proposed by Davidson and Lambert in 2001, constructed the RDST (regional disaster system theory), including a conceptual model for disaster risk assessment [54]. In risk analysis, it is common to build an indicator system based on this model (Equation (1)) [55]. This equation is not a true mathematical formula, and researchers use it to construct a system of metrics and, based on their specific requirements, determine whether the symbol "*" represents multiplication or addition to perform numerical calculations. In our work, addition is used.

$$R = H^{\alpha} * E^{\beta} * V^{\delta} \tag{1}$$

where *R* represents the risk of natural disasters, *H* signifies the hazard factors, *E* represents the hazard-formative environment, and *V* denotes the vulnerability of the hazard-affected body. *α*, *β*, and *δ* represent the weights of the three evaluation factors *H*, *E*, and *V*, respectively, and were calculated based on the respective evaluation indicators chosen [54].

### 2.3.2. Selection of Forest Fire Risk Indicators Based on the RDST

The selection and application of indicators directly affect the accuracy of the forest fire risk assessment results. Based on the natural disaster risk theory previously discussed, we have chosen primary influencing factors from three aspects: hazard factor, hazard-affected body, and hazard-formative environments. We have analyzed their relationships with forest fires and categorized the indicators based on the analysis results.

- Hazard factor.

Referring to the "Forest Fire Danger Weather Ratings" (GB/T 36743-2018) proposed by the China Meteorological Administration in 2018, we calculated the FFDI using several indicators, such as the daily maximum temperature, average 2 min wind speed, minimum relative humidity, drought level, the precipitation correction coefficient, and the snowfall correction coefficient [49]. The FFDI is one of the indicators included in the calculation of the CFFRI. Lightning ignition is the primary natural ignition source for forest fires in Sichuan Province, accounting for approximately 2% of identified forest fire ignition causes, with the remaining 98% attributed to human factors [56,57]. Lightning can present a more impactful role compared to human activities, especially in rural regions, which is appropriated for our study area [18]. Therefore, incorporating lightning data into forest fire risk assessments can partially reflect the impact of natural ignition sources on forest fires.

- The hazard-formative environment.

Different human habitats have varying impacts on forest fires. Areas with dense human activity are more likely to experience forest fires. Locations with a high roading density can effectively impede the spread of forest fires, while areas with poor forest integrity are less prone to forest fires. Hence, we used specific indicators, such as population density and roading density, to represent the influence of human habitats on forest fires within the hazard-formative environment. Bodies of water, such as rivers, can act as barriers to prevent the spread of forest fires and increase air humidity [58]. Terrain factors directly affect the occurrence and development of forest fires. In this context, we considered elevation as a terrain factor, which is generally inversely related to the occurrence of forest fires.

- The hazard-affected body.

The hazard-affected bodies usually refer to administrative units, and their flammability is often calculated based on social statistic indicators [54]. The vulnerability of hazard-affected bodies refers to the degree of vulnerability when affected by hazard factors; a higher vulnerability corresponds to a higher risk [54]. Our hazard-affected body was the forest, and the vegetation categories differed in their combustibility; hence, we selected the vegetation category as a factor to represent its vulnerability [59]. Forests of the study area were categorized into five classes and assigned different vulnerability values based on the following correlation analysis [58].

### 2.3.3. Exploration of the Relationship between Forest Fires and Their Indicators

To ascertain the significance of the relationship between various factors and fire points, Spearman correlation analysis was conducted between the fire risk index and fire point data (Equation (2)). Scatter plots were generated between the logarithmically transformed fire point data and each factor:

$$r_{xy} = \frac{n\sum x_i y_i - \sum x_i \sum y_i}{\sqrt{n\sum x_i^2 - (\sum x_i)^2}\sqrt{n\sum y_i^2 - (\sum y_i)^2}} \quad (2)$$

where $r_{xy}$ represents the Pearson correlation coefficient between variables $x$ and $y$; $n$ stands for the total number of observed data points; $x_i$ denotes the ith observed value of variable $x$; $y_i$ signifies the ith observed value of variable $y$.

The calculation method for the FFDI was derived from the "Forest Fire Weather Ratings" (GB/T 36743-2018) introduced by the China Meteorological Administration, which was implemented in April 2019 (Equations (3) and (4)) [49]. In this dataset, the average 2 min wind speed represents the wind speed at 14:00; the daily maximum temperature represents the temperature at 14:00; the daily minimum relative humidity represents the relative humidity at 14:00, and the 20–20 precipitation was used as the base data for the drought-level calculations, referencing the China Meteorological Data Network's China Ground Climate Data Daily Value Dataset (V3.0) [60]:

$$I_{FFDI} = U * C_r * C_s \tag{3}$$

$$U = \mathrm{f}(V) + \mathrm{f}(T) + \mathrm{f}(r_{RH}) + \mathrm{f}(M) \tag{4}$$

where $I_{FFDI}$ is the value of the FFDI; $U$ stands for the function describing the FFDI; $C_r$ represents the precipitation correction coefficient. When the 24 h precipitation ($R_r$) was greater than or equal to 1 mm, $C_r$ was assigned the value 0; otherwise, it was assigned the value 1; $C_s$ denotes the snowfall correction coefficient. When the 24 h snow depth ($H_s$) was greater than 0 cm, $C_r$ was set to 0; otherwise, it was set to 1. The threshold values for $R_r$ and $H_s$ can be adjusted based on specific climatic conditions [60]; $V$ represents the 14 h wind speed; $T$ stands for the 14 h air temperature; $r_{RH}$ represents the 14 h relative humidity; $M$ represents the drought level. The actual f-function of each of the variables is different, transforming the raw values into dimensionless exponents, which can be added together to obtain $U$, and it's simplified into a function value check table, allowing us to directly determine the function value based on the value range (Tables 2–5).

**Table 2.** Wind speed and its corresponding function value check table.

| $V$/(m/s) | ≤1.5 | (1.5,3.5] | (3.5,5.6] | (5.6,8.1] | (8.1,10.9] | (10.9,14.9] | (14.9,17.2] | >17.2 |
|---|---|---|---|---|---|---|---|---|
| f($V$)/% | 4 | 8 | 12 | 15 | 19 | 23 | 27 | 31 |

**Table 3.** Temperature and its corresponding function value check table.

| $T$/°C | ≤5 | (5,10] | (10,15] | (15,20] | (20,25] | >25 |
|---|---|---|---|---|---|---|
| f($T$)/% | 0 | 5 | 6 | 9 | 13 | 15 |

**Table 4.** Relative humidity and its corresponding function value check table.

| $r_{RH}$/% | ≥70 | [60,70) | [50,60) | [40,50) | [30,40) | <30 |
|---|---|---|---|---|---|---|
| f($r_{RH}$)/% | 0 | 3 | 6 | 9 | 12 | 15 |

**Table 5.** Drought level and its corresponding function value check table.

| M/d | Mild drought No drought | 0 ≤3 | 1 (3,6] | 2 (6,9] | 3 (9,12] | 4 (12,14] | 5 (14,16] | 6 (16,18] | 7 (18,20] | ≥8 >20 |
|---|---|---|---|---|---|---|---|---|---|---|
| f(M)/% | | 0 | 8 | 12 | 19 | 23 | 27 | 31 | 35 | 38 |

### 2.3.4. Fire Risk Indicator Weighting

The measurement of weights represents a physical quantification of the contribution rates of various indicator variables to their respective goals. Common methods for determining weights include subjective and objective approaches, such as the Delphi method, the entropy method, principal component analysis, and the AHP.

Based on the analysis of fire risk indicators, we comprehensively considered the characteristics and major advantages and disadvantages of commonly used weight assignment methods. Taking both subjective and objective aspects into account, we selected the AHP and entropy methods for weight calculations, and then computed their minimum relative entropy to determine the combined weights.

- The analytic hierarchy process.

The AHP allows decision makers to quantitatively incorporate their experiences and judgments into the decision-making process. This method uses the judgment matrix formula to determine the importance levels of different evaluation indicators for the same factor, thereby assesses the impact of evaluation indicators on the results in the final plan. The steps are as follows [61]:

1. Constructing a hierarchy structure model.

Divide decision goals, decision factors (decision criteria), and decision objects according to their mutual relationships into goal layers, criterion layers, and scheme layers (decision objects), respectively.

2. Constructing judgment (pairwise comparison) matrices.

With the consistent matrix method, compare two unrelated factors as much as possible to minimize the impact of factors with different natures. The result of comparing the importance of factor *i* with factor *j* is denoted as $a_{ij}$ [62].

$$a_{ij} = 1/a_{ji} \tag{5}$$

3. Hierarchical single sorting and the consistency test.

The characteristic vector of the maximum eigenvalue, $\lambda_{max}$, of the judgment matrix, after normalization, is known as hierarchical single sorting [63,64]. For a positive reciprocal matrix, *a* of order *n*, the maximum eigenvalue λ is greater than or equal to *n* if and only if λ = *n*, making a consistent matrix [61].

4. Hierarchical overall sorting and the consistency test.

The calculation of the weights representing the relative importance of all factors at a certain level regarding the highest level (the objective level) is referred to as hierarchical overall sorting. This process is carried out sequentially from the highest level to the lowest level [64].

Based on the construction of the CFFRI (composite forest fire risk index) indicator system, we established a hierarchical structure model with the CFFRI as the objective level, the hazard factor, the hazard-formative environment, and the hazard-affected body as the criterion level, and various influencing factors as the indicator level. We conducted a quantitative analysis between each pair of factors within the same objective level and under the same criterion to construct new judgment matrices, ultimately calculating the weight of each judgment indicator [65].

- The entropy method.

The entropy method is an objective weighting method that determines weights based on the objective information reflected by various indicators. It can measure the uncertainty of an event and eliminate the influence of human factors [66]. The smaller the information entropy obtained using the entropy method for each indicator, the lower the disorderliness of the information, and the greater the utility value of the information, leading to a larger weight for the indicator [65,66]. The calculation steps are as follows [67]:

Assuming there are *n* samples and *m* indicators, the value of the *j*-th indicator for the *i*-th sample is denoted as $x_{ij}$ (*i* = 1, 2, 3... *n*; *j* = 1, 2, 3... *m*).

1. Normalize the original data to eliminate the influence of physical quantities.

2. Calculate the proportion $p_{ij}$ of the *i*-th sample under the *j*-th indicator:

$$p_{ij} = x_{ij} / \sum_{i=1}^{n} x_{ij} \qquad (6)$$

3. Calculate the information entropy, $e_j$, for the *j*-th indicator:

$$e_j = \left( -\frac{1}{\ln n} \right) \sum_{i=1}^{n} p_{ij} \ln \left( p_{ij} \right), 0 \leq e_j \leq 1 \qquad (7)$$

4. Calculate the differentiation coefficient:

The differentiation coefficient, $g_j$, also known as the information utility value, is primarily determined through the difference between the indicator's information entropy, $e_j$, and 1.

$$g_j = 1 - e_j \qquad (8)$$

5. Determine the weight, $W_j$, of the evaluation indicator:

$$W_j = \frac{g_j}{\sum_{i=1}^{m} g_j}, j = 1, 2, 3, \dots m \qquad (9)$$

- Combined weight.

The indicator weights calculated using the AHP are represented as $w_{1i}$, and the weights calculated using the entropy method are represented as $w_{2i}$. Based on the principle of minimum relative entropy, the combined weight calculation formula can be obtained using the Lagrange multiplier method [67]:

$$w_i = \frac{\sqrt{(w_{1i} * w_{2i})}}{\sum_{i=1}^{m} \sqrt{(w_{1i} * w_{2i})}}, i = 1, 2, 3, \dots, m \qquad (10)$$

2.3.5. The CFFRI Construction

- Structure designing.

Drawing from the RDST, we computed the CFFRI by considering three key facets: the hazard factor, the hazard-formative environment, and the hazard-affected body. Specifically, we evaluated seven indicators, namely the FFDI, lightning intensity, the vegetation category, population density, distance to rivers, roading density, and altitude, and subsequently standardized the data. Weightings for these indicators were determined through both the AHP and the entropy method, leveraging the concept of minimum relative information entropy. Finally, utilizing the combined weights and the disaster risk assessment model (Equation (11)), we calculated the CFFRI and built the final CFFRI for computing the forest fire risk:

$$R = H^{\alpha} * V^{\beta} * E^{\delta} = (I^{\varepsilon\alpha} * l^{\gamma\alpha}) * \left( t^{\mu\beta} \right) * \left( p^{\upsilon\delta} * r^{\omega\delta} * o^{\chi\delta} * d^{\kappa\delta} \right) \qquad (11)$$

where $H$, $V$, and $E$ represent hazard factors, the hazard-formative environment, and the hazard-affected body, respectively. $\alpha$, $\beta$, and $\delta$ represent the weights for each corresponding factor. $I$, $l$, $t$, $p$, $r$, $o$, and $d$ represent $I_{FFDI}$ (Equation (3)), lightning intensity, the vegetation category, population density, distance to rivers, roading density, and altitude, respectively, while $\varepsilon$, $\gamma$, $\mu$, $\upsilon$, $\omega$, $\chi$, and $\kappa$ represent the corresponding weights for seven indicators:

$$R = (I * \varepsilon\alpha + l * \gamma\alpha) + (t * \mu\beta) + (p * \upsilon\delta + r * \omega\delta + o * \chi\delta + d * \kappa\delta) \qquad (12)$$

where $\varepsilon\alpha$, $\gamma\alpha$, $\mu\beta$, $\upsilon\delta$, $\omega\delta$, $\chi\delta$, and $\kappa\delta$ represent the combination weights obtained through the AHP and entropy method calculations.

- Grading for results.

We utilized the nature breaks classification method to reduce intraclass differences and enhance interclass distinctions. In accordance with practical considerations, we categorized the fire risk values into five levels:

$$SSD_{i-j} = \sum_{k=i}^{j} \left( A[k] - mean_{i-j} \right)^2, 1 \le i < j \le N \tag{13}$$

where $A$ represents an array (with a length of N), and $mean_{i-j}$ signifies the mean value within each level.

## 3. Results

### 3.1. Relation between Fire and Its Indicators

The Spearman method revealed significant correlations between the fire points and the indicators, except for lightning intensity and roading density (Table 6). To further explore and visually represent the relationships, we plotted scatter graphs of cumulative fire points within kilometer grids against each factor, except for the vegetation category (Figure 4a–f). Due to the magnitude of fire point counts, we applied a logarithmic transformation. For the vegetation category, we created histograms and conducted an analysis (Figure 4g).

**Table 6.** Spearman correlation between each indicator and their fire points.

| Indicator | Correlation | Fire Points |
|---|---|---|
| The FFDI | Spearman correlation | 0.237 ** |
| | Sig. (2-tailed) | 0.000 |
| Lightning intensity | Spearman correlation | 0.008 |
| | Sig. (2-tailed) | 0.102 |
| Vegetation category | Spearman correlation | −0.018 ** |
| | Sig. (2-tailed) | 0.000 |
| Population density | Spearman correlation | 0.072 ** |
| | Sig. (2-tailed) | 0.000 |
| Distance to rivers | Spearman correlation | −0.022 ** |
| | Sig. (2-tailed) | 0.000 |
| Roading density | Spearman correlation | -0.005 |
| | Sig. (2-tailed) | 0.395 |
| Altitude | Spearman correlation | −0.163 ** |
| | Sig. (2-tailed) | 0.000 |

**—Correlation is significant at the 0.01 level.

These results indicate that as the FFDI increases, the fire points become denser, with the natural logarithm of fire point count also increasing. Specifically, higher FFDI values were associated with a greater likelihood of forest fires and more fire points. When the lightning intensity was set at 0, there were still a number of fire points present. Between the lightning intensities of 3 kA and 45 kA, there was a substantial number of fire points, with a relatively stable relationship between their natural logarithms. In the 45–90 kA range, both the number of fire points and their natural logarithms decreased. As the lightning intensity continued to rise, both the fire points and their natural logarithms decreased even further.

Regarding population density, as it gradually increases from 0, the corresponding natural logarithm of fire points initially remains high but starts to decline after reaching 300 people/km$^2$. When the distance to rivers was too great, there were very few fire points. However, when the distance exceeded 6000 m, both the number of fire points and their natural logarithms decreased as the distance to rivers increased. When the distance was less than 6000 m, the fire point density increases with increasing distance from the rivers.

For roading density, when it exceeded 3 km per kilometer grid, there were very few fire points. When the roading density exceeded 2 km, both the number of fire points and their natural logarithms decreased. When the roading density was less than 1.5 km, there was no significant change in the relationship between the roading density and both the number of fire points and their natural logarithms. For altitudes below 1000 m, there were relatively few fire points. Between 1000 m and 1500 m, the number of fire points

significantly increased. The fire points peaked between 1500 m and 2500 m, after which the natural logarithm of fire points noticeably decreased. Beyond 3500 m, the fire points decreased with increasing altitude.

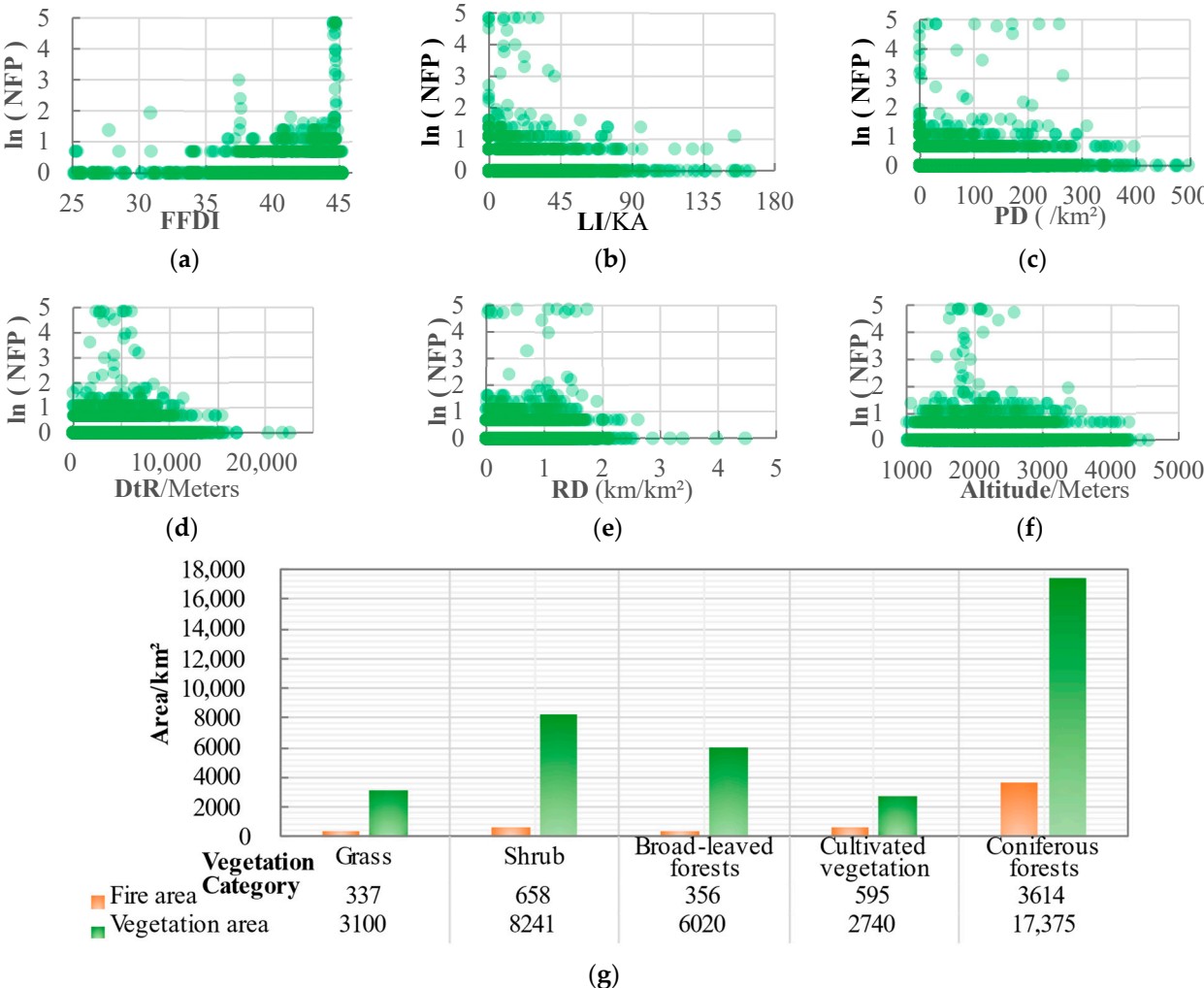

**Figure 4.** Visualization of the NFP and indicators: (**a**–**f**) scatter diagram of cumulative logarithmic NFP and each indicator; (**g**) statistical results between the fire area and different vegetation categories. LI: lightning intensity; PD: population density; DtR: distance to rivers; RD: roading density; NFP: number of fire points.

Regarding the vegetation category, areas with coniferous forests had the largest fire point coverage, approximately 3614 km$^2$ (10-year cumulative), representing 65% of the total fire point area, significantly surpassing the 46.3% forest area covered by coniferous forests. When the vegetation categories were shrubland, broadleaf forests, and grassland, their fire point coverage percentages were 11.8%, 6.4%, and 6.1%, respectively, all significantly lower than their respective proportions in the forest area. Cultivated vegetation areas accounted for 10.7% of the total fire point area, exceeding the 7.3% of the forest area occupied by cultivated vegetation.

*3.2. Grading of Indicators*

Based on the relationship between the fire risk indicators and fire occurrences and utilizing the data from all kilometer grid cells from 2010 to 2019 within the study area, we established a classification standard for forest fire risk levels (Table 7). This classification ranges from one to ten [58,68,69], referencing the relevant literature and exploration of different indicators.

**Table 7.** Grading of forest fire risk indicators.

| The FFDI | ≤25 | (25,30] | (30,35] | (35,40] | >40 |
|---|---|---|---|---|---|
| Fire danger class | 1 | 3 | 6 | 8 | 10 |
| LI * (ka) | ≤3 | (3,45] | (45,90] | (90,135] | >135 |
| Fire danger class | 4 | 8 | 6 | 5 | 4 |
| PD * (/km$^2$) | ≤10 | (10,150] | (150,300] | (300,400] | >400 |
| Fire danger class | 5 | 9 | 7 | 5 | 3 |
| DtR * (km) | ≤1 | (1,3] | (3,6] | (6,12] | >12 |
| Fire danger class | 4 | 7 | 9 | 6 | 5 |
| RD * (km/km$^2$) | ≤1.5 | (1.5,2] | (2,2.5] | (2.5,3] | >3 |
| Fire danger class | 9 | 7 | 5 | 3 | 2 |
| Altitude (m) | ≤1000 | (1000,1500] | (1500,2500] | (2500,3500] | >3500 |
| Fire danger class | 5 | 8 | 10 | 7 | 4 |
| VC * | Coniferous forests | Cultivated vegetation | Broad-leaved forests | Shrub | Grass |
| Fire danger class | 10 | 9 | 7 | 7 | 5 |

* LI: lightning intensity; PD: population density; DtR: distance to rivers; RD: roading density; VC: vegetation category.

### 3.3. Construction of the CFFRI

### 3.3.1. Subjective Weight Analysis for the Indicator System

We investigated the pairwise importance of hazard factors, the hazard-formative environment, and the hazard-affected body with the AHP in terms of the quantity and characteristics of fire indicators (Table 8). Meteorological factors are the primary factors influencing forest fire occurrences, making hazard factors the most crucial. Following that are the hazard-formative environment and vulnerable elements. The weightings for these three factors were determined through the construction of the judgment matrix, resulting in weights of 0.575, 0.343, and 0.082, respectively. Utilizing the AHP for hierarchical ranking and consistency testing of the matrix model data, when the order of the matrix was three, RI = 0.58, and CR < 0.1, meeting the consistency testing requirements.

**Table 8.** Judgment matrix and weight of criterion importance.

| Criterion Layer | H * | E * | V * | Weight | Consistency |
|---|---|---|---|---|---|
| H * | 1 | 2 | 6 | 0.575 | |
| E * | 1/2 | 1 | 5 | 0.343 | $\lambda_{max} = 3.029$ |
| V * | 1/6 | 1/5 | 1 | 0.082 | CR = 0.025 < 0.1 |

* H: the hazard factor; E: the hazard-formative environment; V: vulnerability of the hazard-affected body.

Subsequently, based on the hazard factor criterion, we explored the importance of the FFDI and lightning intensity. Since the FFDI can better represent its relationship with forest fires, while lightning intensity contributes relatively less to the ignition of forest fires, we assigned weights of 0.90 to the FFDI and 0.10 to lightning intensity when constructing the judgment matrix (Table 9). This reflects the significant impact of the FFDI on the CFFRI compared to lightning intensity. These weightings have been tested and found to meet the requirements of hierarchical ranking and consistency testing.

**Table 9.** Judgment matrix and weights between the hazard factor, lightning intensity, and the FFDI.

| H * | FFDI * | LI * | Weight | Consistency |
|---|---|---|---|---|
| FFDI * | 1 | 9 | 0.900 | $\lambda_{max} = 2.000$ |
| LI * | 1/9 | 1 | 0.100 | CR = 0.000 < 0.1 |

* H: the hazard factor; FFDI: the forest danger index; LI: lightning intensity.

Finally, based on the hazard-formative environment criterion, we explored the relative importance of population density, distance to rivers, roading density, and altitude. From

the relationships between these factors and the number of fire points, it was evident that these four factors do not exhibit purely positive or negative effects on the fire point counts (Table 10). There existed an intermediate value with a relatively significant influence. However, the population density, distance to rivers, and altitude showed significant correlations with the fire points. Taking into account the above analysis and practical considerations, we constructed a judgment matrix to determine the weights as follows: population density 0.365, distance to rivers 0.147, roading density 0.097, and altitude 0.391. When the matrix order was four, RI = 0.9, and CR < 0.1, meeting the consistency testing requirements.

**Table 10.** Judgment matrix and weights between the hazard-formative environment and indicators.

| E * | PD * | DtR * | RD * | Altitude | Weight | Consistency |
|-----|------|-------|------|----------|--------|-------------|
| PD * | 1 | 3 | 3 | 1 | 0.365 | |
| DtR * | 1/3 | 1 | 2 | 1/3 | 0.147 | $\lambda_{max} = 4.046$ |
| RD * | 1/3 | 1/2 | 1 | 1/4 | 0.097 | $CR = 0.017 < 0.1$ |
| Altitude | 1 | 3 | 4 | 1 | 0.391 | |

* E: the hazard-formative environment; PD: population density; DtR: distance to rivers; RD: roading density.

Among all indicators within the three criteria, the FFDI had the most significant influence on the results, accounting for more than half at 51.8%. The altitude and population density closely followed, with their impact on the CFFRI being relatively equal, constituting from 12% to 14% of the total impact. The vegetation category came next, contributing 8.2%. The lightning intensity, distance to rivers, and roading density each accounted for 5.8%, 5%, and 3.3%, respectively (Table 11).

**Table 11.** The weight of each indicator calculated using the AHP.

| Target Layer | Criteria Layer * | Criteria Weight | Index Layer * | Index Weight | Final Weight |
|--------------|------------------|-----------------|---------------|--------------|--------------|
| | H | 0.575 | FFDI | 0.900 | 0.518 |
| | | | LI | 0.100 | 0.058 |
| CFFRI | E | 0.343 | PD | 0.365 | 0.125 |
| | | | DtR | 0.147 | 0.050 |
| | | | RD | 0.097 | 0.033 |
| | | | Altitude | 0.391 | 0.134 |
| | V | 0.082 | VC | 1.000 | 0.082 |

* H: the hazard factor; E: the hazard-formative environment; V: vulnerability of the hazard-affected body; LI: lightning intensity; PD: population density; DtR: distance to rivers; RD: roading density; VC: vegetation category.

### 3.3.2. Objective Weight Calculations of the Fire Indicators

Due to significant differences in the data of various indicators, conducting dimensionless processing on the raw data would result in substantial variations in the calculation results. Therefore, we employed the entropy method for weight calculations based on the pre-established risk level divisions for each indicator, with all risk levels ranging from one to ten. This approach allowed us to reduce numerical differences while preserving the characteristics of the data. The specific weight distributions were as follows: the FFDI and LI each contributed approximately 20%, PD and altitude around 15%, RD had a lower weight of 6%, and DtR and VC accounted for approximately 11% (Table 12). This weighting reflects that the FFDI and LI have relatively lower entropy levels, indicating lower disorder in the information, higher differential coefficients, and greater utility of information. Consequently, they received higher weights, while the other indicators followed in descending order.

**Table 12.** The weight of each indicator calculated using the entropy method.

| Indicator * | Information Entropy | Difference Coefficient | Weight |
|---|---|---|---|
| The FFDI | 0.995 | 0.005 | 0.208 |
| LI | 0.995 | 0.005 | 0.210 |
| PD | 0.996 | 0.004 | 0.154 |
| DtR | 0.997 | 0.003 | 0.112 |
| RD | 0.999 | 0.001 | 0.057 |
| Altitude | 0.997 | 0.003 | 0.148 |
| VC | 0.997 | 0.003 | 0.111 |

* LI: lightning intensity; PD: population density; DtR: distance to rivers; RD: roading density; VC: vegetation category.

3.3.3. Weight Combination for the CFFRI

Based on the results of the two weight calculation methods, it was evident that the weights calculated using the entropy method are more balanced than those obtained through the AHP (Figure 5). Notably, the indicators with significant differences in the results between the two methods are the FFDI and lightning intensity The former calculates objective weights based on the inherent differences in each type of data, while the latter involves subjective judgments and quantitative-level divisions by the researcher when calculating the weights. Both approaches have their advantages and drawbacks.

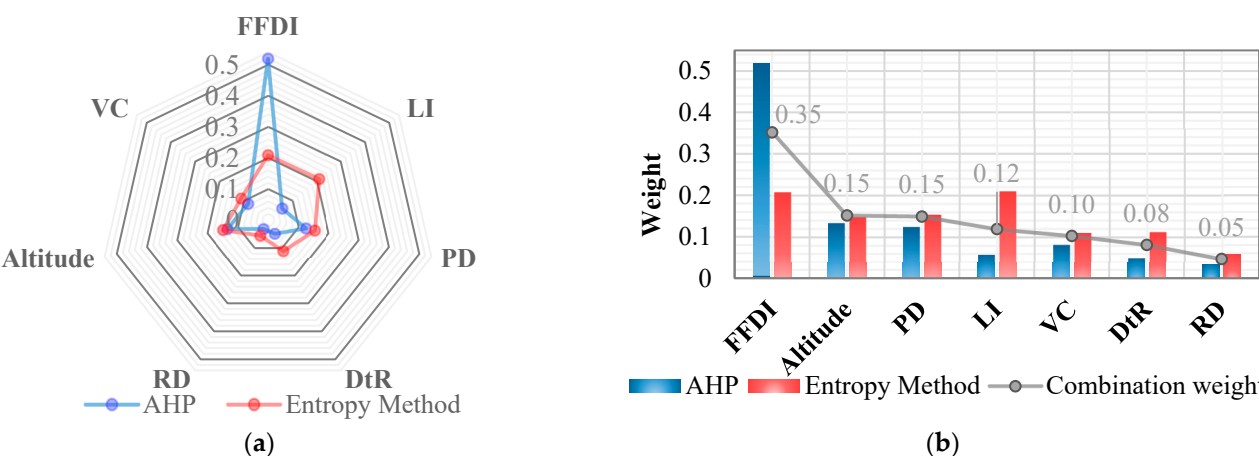

**Figure 5.** Weight values of the AHP, the entropy method (**a**), and the combination results (**b**); LI: lightning intensity; PD: population density; DtR: distance to rivers; RD: roading density; NFP: number of fire points.

Subsequently, employing the principles of minimum relative entropy and the Lagrange multiplier method, we further computed the combined weights for the seven indicators based on the results of the AHP and the entropy method. These final weights were then incorporated into the calculation of the CFFRI:

$$R = 0.35I + 0.12l + 0.10t + 0.15p + 0.08r + 0.05o + 0.15d \tag{14}$$

where $I$ and $l$ represent the hazard factors, namely the FFDI and lightning intensity, respectively; $t$ represents the vulnerable element, which is the vegetation category; $p$, $r$, $o$, and $d$ stand for elements within the hazard-formative environment, representing the population density, distance to rivers, roading density, and altitude, respectively.

*3.4. Forest Fire Risk from 2010 to 2019*

According to the CFFRI, we utilized the assigned risk levels for the seven indicators to calculate the forest fire risk values in the study area for the years 2010–2019, with the risk scale ranging from zero to ten. Employing the natural breaks classification method, we established a classification table for the risk index (Table 13). Based on this risk level

table, we generated a fire risk distribution map for the study area (Figure 6). High-risk areas were predominantly located in the central and southern parts of the study area, while moderately high-risk areas were mainly situated in the southern and southwestern regions. Medium-risk zones were prevalent in the eastern part of the moderately high-risk area. Moderately low-risk areas were primarily found in the northern and eastern sections of the study area, while low-risk areas were concentrated in the northeastern part. These different fire risk zones exhibited distinct boundaries.

**Table 13.** Classification table for the CFFRI.

| The CFFRI | Fire Risk Class | Fire Risk Description |
|---|---|---|
| ≤4.7 | I | Low risk |
| (4.7,5.3] | II | Mid-low risk |
| (5.3,6.5] | III | Medium risk |
| (6.5.7.6] | IV | Mid-high risk |
| >7.6 | V | High risk |

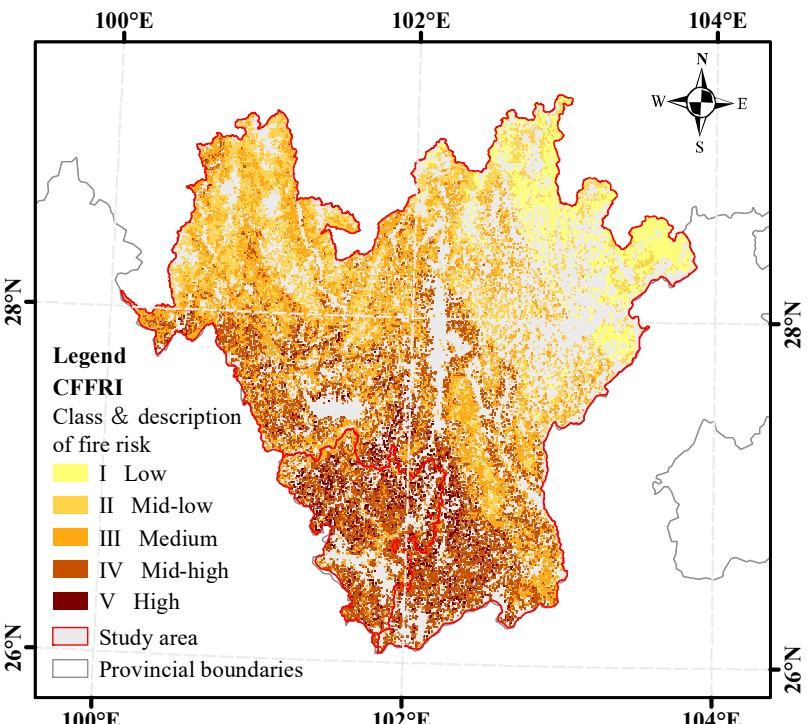

**Figure 6.** Forest fire risk of Liangshan Yi autonomous prefecture and Panzhihua from 2010 to 2019.

*3.5. Comparison and Validation between the CFFRI and the FFDI*

We selected historical forest fire cases with relatively large burned areas and closely located fire incidents (Table 14). The specific validation procedure involved calculating the average CFFRI from the start to the end of each wildfire event. This average CFFRI value was then compared to areas surrounding the fire-affected zone and also compared to the FFDI results. This comparison allowed us to determine whether the average CFFRI during the wildfire period was in a significantly higher state.

To ensure the accuracy of the fire risk location information, we needed to compare the fire risk index calculation results with the MCD64 fire data for the month of the fire incident. The data processing steps were as follows:

1. Calculate the FFDI for selected forest fire events and perform spatial interpolation.
2. Divide the interpolation results into five risk levels based on FFDI standards (Table 15), which facilitates the comparison with the CFFRI.

3. Utilize the FFDI level values to compute the daily CFFRI values.
4. Determine the average FFDI and fire risk level, as well as the average CFFRI and risk level during the occurrence of the wildfires.
5. Compare the village coordinates with the MCD64 fire data to confirm the actual fire locations (Figure 7). Compare the fire risk values at these locations with the areas unaffected by fires to assess the accuracy of the two calculation methods in relation to the actual fire incidents.

**Table 14.** Validation and comparison cases of the CFFRI and the FFDI (area unit: hectare).

| Start Time | End Time | Burned Area | Cause of Fire | Longitude | Latitude |
|---|---|---|---|---|---|
| 29 January 2010 | 5 February 2010 | 128 | Smoking | 101.55 | 28.16 |
| 4 April 2013 | 5 April 2013 | 10 | - | 101.74 | 26.50 |
| 5 April 2013 | 5 April 2013 | 10 | - | 102.10 | 26.88 |
| 12 February 2014 | 16 February 2014 | 23 | - | 100.91 | 28.49 |
| 14 February 2014 | 16 February 2014 | 56 | - | 102.22 | 28.57 |
| 13 April 2014 | 14 April 2014 | 70 | - | 101.65 | 26.56 |
| 13 April 2014 | 14 April 2014 | 11 | Smoking | 102.28 | 28.36 |
| 15 April 2014 | 18 April 2014 | 69 | - | 102.33 | 28.35 |
| 19 March 2016 | 22 March 2016 | 81 | Ancestor worship | 102.20 | 28.34 |

**Table 15.** Grading for the FFDI.

| FFDI | [4,38) | [38,43) | [43,65) | [65,73) | [73,100) |
|---|---|---|---|---|---|
| Grading | 1 | 3 | 5 | 7 | 9 |

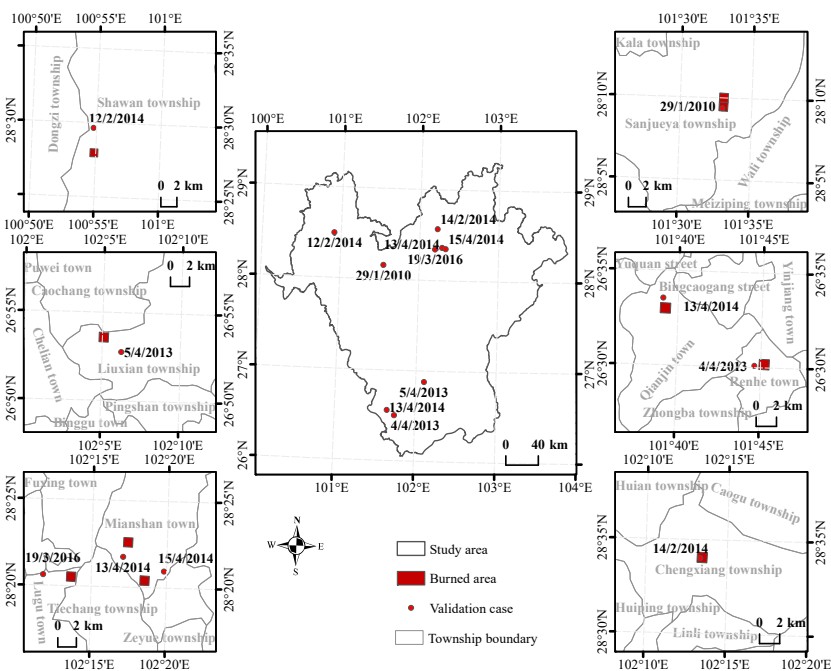

**Figure 7.** Validation cases with burned areas of MCD64.

The results obtained from the CFFRI indicate that the overall fire risk values were relatively high, with 89% falling into the high-risk category, and 11% in the medium-risk category (Table 16). This outcome is more consistent with the actual fire incidents compared to the FFDI, where the IV fire risk comprises 56% and the III fire risk makes up 44% of the total. To assess the accuracy of the fire risk calculation, we compared the fire risk values of burned pixels to those of unburned pixels. We calculated the probability of burned

pixels having higher fire risk values than the surrounding unburned pixels. A higher probability indicates a more accurate fire risk calculation [69]. Figure 8 illustrates that the CFFRI calculation results are predominantly positioned above the FFDI calculation results, suggesting that the CFFRI yields more instances where the fire risk values are higher than the surrounding pixel values, thereby achieving a higher level of accuracy. The calculated accuracy of the CFFRI results was 85%, while the FFDI results had an accuracy of 76%. This demonstrates that our proposed model, within the context of the data and calculation methods used in this study, outperforms the traditional FFDI. It further underscores the applicability of the CFFRI in this region.

**Table 16.** Accuracy comparison between the CFFRI and the FFDI.

| Fire Case | | CFFRI | | | FFDI | | |
|---|---|---|---|---|---|---|---|
| | | Value | Class | Accuracy | Value | Class | Accuracy |
| 1 | 29 January 2010 Liangshan | 7.18 | High | 100% | 70.72 | IV | 80% |
| 2 | 4 April 2013 Panzhihua | 7.03 | High | 100% | 71.74 | IV | 88% |
| 3 | 5 April 2013 Panzhihua | 6.90 | High | 75% | 66.39 | IV | 75% |
| 4 | 12 February 2014 Liangshan | 6.52 | High | 100% | 55.25 | III | 63% |
| 5 | 12 February 2014 Liangshan | 6.31 | Medium | 50% | 61.87 | III | 75% |
| 6 | 13 April 2014 Panzhihua | 6.94 | High | 88% | 71.92 | IV | 63% |
| 7 | 13 April 2014 Liangshan | 7.13 | High | 100% | 59.95 | III | 88% |
| 8 | 15 April 2014 Liangshan | 7.01 | High | 50% | 70.8 | IV | 50% |
| 9 | 19 March 2016 Liangshan | 6.53 | High | 100% | 53.56 | III | 100% |

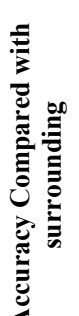
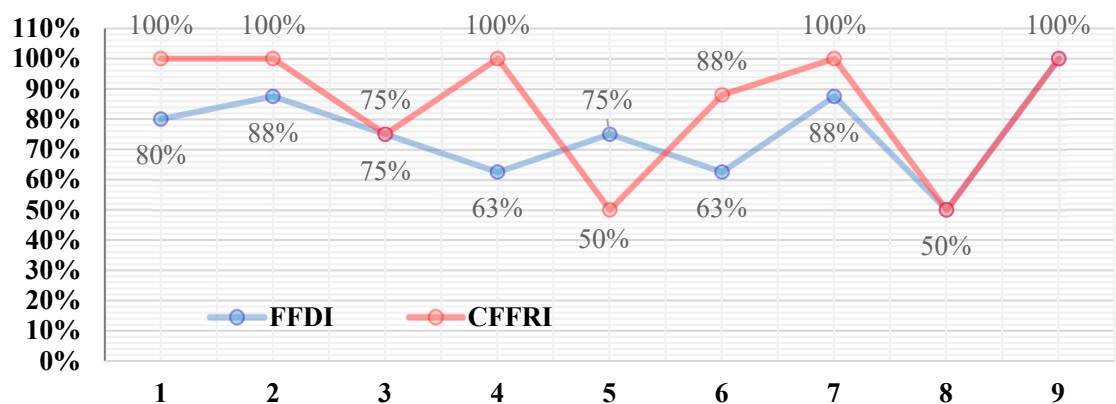

**Figure 8.** Accuracy comparison between the CFFRI and the FFDI.

## 4. Discussion

### 4.1. Correlations between Forest Fire and Its Indicators

Among all indicators in the CFFRI, the FFDI exhibited a noticeable correlation with the fire points and can be considered the primary element in calculating forest fire risk. Lightning intensity showed a relatively weak correlation with the number of fire points, but different levels of lightning intensity have varying effects on the fire points and can be considered a secondary influencing factor. It is important to note that the lack of a significant correlation with the fire points does not necessarily imply that a factor is unimportant. For instance, the lack of a significant correlation between lightning intensity and fire points can be attributed to the relatively small proportion of lightning-induced fires. When lightning intensity is high, the number of fire points may decrease, which may be related to the actual weather conditions, as severe thunderstorms often accompany high lightning intensities, and other factors may come into play when the lightning intensity is high.

Roading density did not display a significant relationship with the fire points, but as its density increased, there was a trend of an initial increase followed by a decrease in the number of fire points. These findings are consistent with previous research, indicating that

more densely populated areas tend to experience a higher frequency of wildfires [70,71]. However, the generalizability of these results to other regions remains to be validated. The insignificant correlation between roading density and fire points may be influenced by other factors, such as frequent human activities in densely connected road networks, which do not necessarily increase or decrease the probability of forest fires. Population density, distance to rivers, and altitude also exhibited similar characteristics.

Previous studies have shown that when the population density is high, there is a power–law relationship between the probability of fire occurrence and population density, following a fitted curve [72]. At lower population densities, deviations from the power–law relationship occur, primarily due to non-human factors, such as spontaneous combustion and lightning. When the population density is low, there is a higher likelihood of forest fires being triggered by other factors. As the population density increases, human-related factors trigger forest fires more, but over a certain threshold, the hazards may decrease, possibly due to relevant policies or an increase in fire awareness.

Generally, for every 100 m increase in altitude, the temperature decreases by 0.6 °C. Therefore, at higher altitudes, the influence on temperature is greater, making forest fires less likely to occur. The study area's altitude ranges from 500 m to 5000 m, with higher overall altitudes and fewer areas below 1000 m, resulting in fewer fire points in this range. Altitudes between 1000 m and 2500 m are in the high-incidence range for local forest fires, with a gradual decline thereafter.

### 4.2. CFFRI Results

In the construction process of the CFFRI, the AHP approach takes a more subjective perspective by giving a considerable weight to the characteristics of each indicator. This is evident in the fact that the FFDI was assigned half of the total weight, even though its weight differs by as much as 0.3 when compared to the entropy method. However, the FFDI remained the dominant factor in both weighting methods, which underscores the significance of the traditional FFDI. In the AHP results, the latitude and population density were also noteworthy, despite their relatively small weights of around 0.1. This aligns with existing research on the influence of human activities on fire incidents, adding credibility to the CFFRI [18]. On the other hand, lightning intensity shows significant disparities in both weighting methods, primarily due to its weaker correlation in the analysis, leading to its reduced importance in the AHP approach. In summary, both methods have their advantages and disadvantages, ultimately contributing to a more balanced and reliable final model.

The CFFRI results graph reveals distinct spatial patterns in the overall fire risk over the past decade. The entire region of Panzhihua City was consistently placed within the high to very high fire risk levels, with other high-risk areas clustered around its periphery, which can serve as valuable references for the local authorities to enhance targeted fire monitoring and prevention efforts. After validation, it was evident that the CFFRI risk values tended to be higher and closer to real-world situations compared to the FFDI. This alignment with real fire incidents was expected, as our validation data were based on actual fire occurrences [73]. This higher risk estimation could potentially improve the accuracy and practical relevance of fire risk assessments, which is consistent with previous research that has highlighted the challenges associated with threshold determination when fire risk model results are either too high or too low [45]. However, whether this elevated risk estimation truly indicates overestimation in risk calculations remains to be verified. Further investigations, such as a comparative analysis of how both methods assess fire risk in unburned pixels, are needed to clarify this aspect.

### 4.3. Reflections and Prospects

In our work, correlation analysis was used to provide a quantitative and visual relationship between forest fire indicators and fire points from 2010 to 2019 for a more meticulous categorization of the indicators, which was directly based on MOD14. Currently,

there is relevant research confirming the feasibility of using MOD14 directly for validating fire risk modeling [74], with some studies indicating an accuracy level of up to 97.5% [75]. Additionally, for this study, the cumulative values of 10 years of MOD14 data were used for analysis, which compensates for the uncertainty issues of a few fire points. Accordingly, in our study, the impact of errors on the results should be minimal, and the correlation between fire points and indicators is acceptable, while the impact of the quality of forest fire data on the results is still worth exploring.

The main indicators selected for this article include seven in total, with the FFDI encompassing six sub-indicators related to meteorological factors. Our consideration of hazard factors has been fairly comprehensive in our study, and all the fire risk indicators we selected are fundamental and representative. It might be worthwhile to consider incorporating more potential factors into the fire risk assessment model, particularly in the hazard-formative environment aspect. This could involve terrain features like the slope, aspect, and surface roughness. The study area is predominantly forested, characterized by relatively low levels of urbanization. In regions with higher economic and social development, the optimization of the model could involve the inclusion of human-related indicators, such as urbanization rates and GDP. For the hazard-affected body, considerations may extend to the NDVI, EVI, LAI, and vegetation products like NPP.

Investigating the relationship between the MOD14 product and the actual occurrence of fire incidents is also a subject for future research. The limited number of cases and relatively small burnt areas in the collected data impose certain constraints when validating our results. This also leaves uncertainty on the impact of the MCD64 products on the results. In the future, it is essential to supplement our research with more comprehensive historical disaster data to enhance the robustness of our findings.

## 5. Conclusions

Building upon the FFDI, we selected seven categories of influencing factors based on the RDST, encompassing hazard factors, the hazard-formative environment, and the hazard-affected body. We analyzed their interrelationships with forest fire occurrence and constructed a forest fire risk assessment index system and a composite forest fire risk index, which is the CFFRI. We computed fire risk from 2010 to 2019 and compared the results with historical disaster data against the traditional FFDI results. The main conclusions contain two aspects:

The FFDI is directly proportional to the number of forest fire points, while the other indicators show an increase in fire point count with increasing indicator values, up to a critical threshold. Beyond the threshold, an increase in indicator values results in a decrease in fire point count.

The CFFRI shows an overall high fire risk, with 89% falling into the high-risk category and 11% in the mid-high category. This is closer to the actual fire situation compared to the FFDI, which has 56% in the high-risk category and 44% in the mid-high category. The CFFRI yields more high-risk values than the surrounding grid cells, with an accuracy rate of 85%, while the FFDI has an accuracy rate of 76%, demonstrating a superior applicability of the CFFRI compared to the traditional FFDI in this region.

**Author Contributions:** Conceptualization, A.G., Y.Y. and Z.H.; methodology, A.G. and L.L.; software, Z.H., W.B. and H.W.; validation, L.L., Z.H. and A.G.; formal analysis, A.G. and Z.H.; investigation, L.L. and Y.Y.; resources, A.G. and L.L.; data curation, Z.H., Y.Y., W.B. and H.W.; writing—original draft preparation, A.G., Z.H., W.B. and H.W.; writing—review and editing, Z.H. and Y.Y.; visualization, Z.H. and Y.Y.; supervision, L.L.; project administration, A.G.; funding acquisition, A.G. and Z.H. All authors have read and agreed to the published version of the manuscript.

**Funding:** This research was funded and supported by the Open Fund of State Key Laboratory of Remote Sensing Science and Beijing Engineering Research Center for Global Land Remote Sensing Products (Grant No. OF202212) and the National Key Research and Development Program of China under Grant (2023YFC3008303).

**Data Availability Statement:** Our research data are from relevant open data websites, which can be obtained according to the links listed in Section 2.2 of this paper.

**Acknowledgments:** The authors would like to thank the high-performance computing support from the Center for Geodata and Analysis, Faculty of Geographical Science, Beijing Normal University. We are also very grateful to the anonymous reviewers for their valuable comments and suggestions for the improvement of this paper.

**Conflicts of Interest:** The authors declare no conflict of interest.

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
