# Peer review of "Development of an Index for Forest Fire Risk Assessment Considering Hazard Factors and the Hazard-Formative Environment"

_remotesensing, doi:10.3390/rs15215077_

Round 1
Reviewer 1 Report
Comments and Suggestions for Authors
1. This article more fully considered the influence of non-meteorological disaster factors, and constructed a more complete risk indicator system based on the Forest Fire Danger Index (FFDI). And the weights are given by a combination of subjective and objective methods such as hierarchical analysis and entropy value method, which avoids the influence on the results caused by the weights given by human subjectivity. Finally, the results are validated with MCD64A1 overfire area data and historical fire data, which are more convincing. The application prospect is broad, and it is of great practical significance for forest fire risk monitoring and early warning.
2. There are a number of issues that must be addressed before publication can be considered. If the following problems are well solved, we believe that the important contribution of this paper is a very important reference for the research related to the construction of forest fire risk assessment index system.
(1)Combustibles are a key factor in forest fires and are not included in this manuscript for forest fire risk considerations, and it is recommended that they be considered. Factors such as wind speed and temperature are also important in determining whether or not a forest fire occurs, but the choice of only lightning in this manuscript is debatable.
(2)Multiple tables in the manuscript are not listed in the body of the text.
(3)Finding 3.1 is partially problematic in the following ways. First, the MOD14A1 fire point data contains three types of fire point information data, low, medium, and high confidence, and the data presentation for this study (Section 2.2.7) does not indicate exactly which confidence level or levels of fire points were used for the study. Secondly, the correlation analysis using MOD14A1 fire point data with each indicator to discuss the influence of each indicator on forest fire risk may not be convincing enough. The accuracy of MOD14A1 fire point data is still to be verified whether it is reliable or not, which is not mentioned in this study, and the results are not convincing.
(4)Queries on part 3.5 of the findings. The results of this study showed that 89% of the areas mapped as fire risk by the expanded assessment system (CFFRI) were high risk, while the FFDI yielded 56% of the areas as high risk. The authors concluded that CFFRI is more accurate than the traditional forest fire danger index (FFDI). Is it possible that this is due to higher validation accuracy caused by the large size of the high-risk areas in the risk maps produced by CFFRI?
(5)The ending does not adequately summarize the article. This study summarizes and analyzes forest fire risk only in terms of the correlation between the indicators and MOD14A1 fire point data. The other results, the limitations of this study, and the outlook for future research are not adequately summarized and analyzed in the conclusion of this study.
(6)The discussion section of the manuscript needs to be further deepened, and it is suggested that the corresponding headings be added.
Author Response
Dear reviewer,
We would like to thank you for your careful reading, helpful comments, and constructive suggestions, which has significantly improved the presentation of our manuscript.
We have carefully considered all comments and revised our manuscript accordingly. The manuscript has also been double-checked, and the typos and grammar errors we found have been corrected. In the attachment file, we summarize our responses to the comments. We hope our revised manuscript can be accepted for publication soon.
The detailed point-by-point response can be found in the attachment file.
Thanks again for your valuable advice.
Best regards,
All authors

Reviewer 2 Report
Comments and Suggestions for Authors
The article is very interesting and presents the Composite Forest Fire Risk Index (CFFRI) and its advantages compared to the available methods for assessment of forest fires.
The study is well prepared and is original. The structure of the manuscript corresponds to the requirements of the journal.
I have the following small remarks and comments:
1. The text under Fig. 1 and Fig. 3 is very basic and can be improved.
2. The conclusion section can be extended with propositions for further improvements and evaluations of the CFFRI, based on different scenarios or locations.
3. The CFFRI is taking into account seven factors. The authors can invest more efforts in the discussion about the selection of these factors and if there are other factors, which might have minor or major impact on the developed index or the fire risk.
Comments on the Quality of English LanguageSome minor language issues were detected in the article (missing commas, improper use of plural/singular forms, use of wrong tenses, etc.).
A detailed proofreading of the article will fix these issues.
Author Response
Dear reviewer,
We would like to thank you for your careful reading, helpful comments, and constructive suggestions, which has significantly improved the presentation of our manuscript.
We have carefully considered all comments and revised our manuscript accordingly. In the attachment file, we summarize our responses to the comments. We hope our revised manuscript can be accepted for publication soon.
The detailed point-by-point response can be found in the attachment file.
Thanks again for your valuable advice.
Best regards,
All authors

Reviewer 3 Report
Comments and Suggestions for Authors
The authors claim to have proposed a new index model called CFFRI (Composite Forest Fire Risk Index) which, according to the experimental results, has been found to be superior to the existing index FFDI (Forest Fire Danger Index). Although this work is quite intriguing, I am supporting the publication of the current manuscript, due to the following reasons.
1. The current manuscript is not well orgonized. Some contents/paragragphs can be compressed because they are common sense, e.g., fire destruction and RS sallites in the introdution section. Instead, the method and the index for the forest fire risk should be further detailed.
2. Causes of forest fires should be included, as they play a significant role in understanding the topic. However, in this current version, they are not addressed. As mentioned in [1,2], there have been significant changes in forest fire causes, particularly in recent years. These include an increase in wildfire incidents, a rise in human activities contributing to fires, and an unclear seasonal pattern, among other factors. It is important to note that these factors may not be the main focus of the discussion, but they should still be mentioned. Furthermore, due to these reasons, vision-based SAG (space-air-ground) RS has emerged as a mainstreaming method for forest fire detection.
[1] Ying, L., Han, J., Du, Y.,et al. Forest fire characteristics in China: Spatial patterns and determinants with thresholds. Forest ecology and management, 2018, 424, 345-354.
[2] Yang, X., Hua, Z., Zhang, L., et al. Preferred Vector Machine for Forest Fire Detection. Pattern Recognition, 2023, 109722.
3. Some descriptions need to be more detailed, such as those regarding data presentation, experiment setting, parameter selection, etc. This is because the journal of remote sensing has a diverse readership, including individuals from various research fields. It is important to provide sufficient information for experimental reproduction, such as making the data used in the manuscript, and even the codes for the solution, accessible to readers. However, in the current manuscript, there are inconsistencies in the data format. For example, in Table 1, the "Meteorological Data" is in CSV format, while the data for "Vegetation type" is provided in TIFF format. It is unclear how these different types of data are organized together for computation of the indexes. Such unclear descriptions also appeared in the models, e.g., for FDDI, what is the function f used in (4) in this work? Likewise, there have so many variables and weights in (11). Responding to the data in Table 1, what are the meanings of these symbols?
4. There are numerous inappropriate references present in the work. For instance, Ref. [33] seems to be an informal research publication, possibly a Master's thesis or PhD dissertation. It is advisable to minimize the use of such materials since they have not undergone thorough review, unlike the papers published in journals or conferences. Moreover, If the paper is written in Chinese, it is important to clarify this in the reference by stating something like "Chinese paper with English abstract".
Comments on the Quality of English LanguageThe manuscript is easily readable.
Author Response

(The authors gave the same response as above.)

Round 2
Reviewer 1 Report
Comments and Suggestions for Authors
The author has made modifications according to my review comments
Reviewer 3 Report
Comments and Suggestions for Authors
Compared to the old manuscript, the current version has undergone significant improvements. All of questions raised by the reviewer have been accurately addressed.
Comments on the Quality of English LanguageTo the readers of remote sensing, it has readability.